# Lantibiotics Produced by Oral Inhabitants as a Trigger for Dysbiosis of Human Intestinal Microbiota

**DOI:** 10.3390/ijms22073343

**Published:** 2021-03-25

**Authors:** Hideo Yonezawa, Mizuho Motegi, Atsushi Oishi, Fuhito Hojo, Seiya Higashi, Eriko Nozaki, Kentaro Oka, Motomichi Takahashi, Takako Osaki, Shigeru Kamiya

**Affiliations:** 1Department of Infectious Diseases, Kyorin University School of Medicine, Tokyo 181-8611, Japan; motomichi.takahashi@miyarisan.com (M.T.); osaki@ks.kyorin-u.ac.jp (T.O.); skamiya@ks.kyorin-u.ac.jp (S.K.); 2Division of Oral Restitution, Department of Pediatric Dentistry, Graduate School, Tokyo Medical and Dental University, Tokyo 113-8510, Japan; doctormizuho@ac.cyberhome.ne.jp (M.M.); oishi.dohs@tmd.ac.jp (A.O.); 3Institute of Laboratory Animals, Graduate School of Medicine, Kyorin University School of Medicine, Tokyo 181-8611, Japan; f-hojo@ks.kyorin-u.ac.jp; 4Central Research Institute, Miyarisan Pharmaceutical Co. Ltd., Tokyo 114-0016, Japan; s.higashi@miyarisan.com (S.H.); k.oka@miyarisan.com (K.O.); 5Core Laboratory for Proteomics and Genomics, Kyorin University School of Medicine, Tokyo 181-8611, Japan; enozaki@ks.kyorin-u.ac.jp

**Keywords:** lantibiotics, Mutacin, Smb, oral bacteria, intestinal microbiota, dysbiosis

## Abstract

Lantibiotics are a type of bacteriocin produced by Gram-positive bacteria and have a wide spectrum of Gram-positive antimicrobial activity. In this study, we determined that Mutacin I/III and Smb (a dipeptide lantibiotic), which are mainly produced by the widespread cariogenic bacterium *Streptococcus mutans*, have strong antimicrobial activities against many of the Gram-positive bacteria which constitute the intestinal microbiota. These lantibiotics also demonstrate resistance to acid and temperature. Based on these features, we predicted that lantibiotics may be able to persist into the intestinal tract maintaining a strong antimicrobial activity, affecting the intestinal microbiota. Saliva and fecal samples from 69 subjects were collected to test this hypothesis and the presence of lantibiotics and the composition of the intestinal microbiota were examined. We demonstrate that subjects possessing lantibiotic-producing bacteria in their oral cavity exhibited a tendency of decreased species richness and have significantly reduced abundance of the phylum Firmicutes in their intestinal microbiota. Similar results were obtained in the fecal microbiota of mice fed with *S. mutans* culture supernatant containing the lantibiotic bacteriocin Mutacin I. These results showed that lantibiotic bacteriocins produced in the oral cavity perturb the intestinal microbiota and suggest that oral bacteria may be one of the causative factors of intestinal microbiota dysbiosis.

## 1. Introduction

The human microbiota is a complex community of microorganisms occupying the specific habitat of the human body [1]. Different microbial communities are formed at each human body site. Amongst them, the intestinal microbiota are the most important bacteria in both quality and quantity and have a critical role in the maintenance of host human health particularly in promoting intestinal metabolism, maturing the immune system, and protecting against pathogenic bacterial colonization and invasion [2]. The intestinal microbiota of each individual are unique; however, it is generally conserved at the phylum level and can be classified into four dominant phyla: Bacteroidetes and Firmicutes are most dominant followed by Proteobacteria and Actinobacteria [3]. The majority of Firmicutes and Actinobacteria consist of Gram-positive bacteria that include commensal organisms as well as pathogens.

Imbalance in the gut microbiota, termed dysbiosis, is usually characterized by change in the human microbiota from a healthy to a disease pattern [4]. Alternations in the intestinal microbiota can result from exposure to various environmental factors, such as diet, toxins, drugs, and pathogens [5]. There is accumulated evidence that the dysbiosis of the intestinal microbiota may drive infection [6], obesity [7,8], and diabetes [9,10]. For instance, it was reported that in the intestinal microbiota of obese people, the proportion of Firmicutes was increased, and that of Bacteroidetes was decreased [11].

Bacteriocins of Gram-positive bacteria are divided into four classes and lantibiotics belong to class I. Lantibiotics are ribosomally synthesized and posttranslationally modified with amino acids such as dehydroalanine (Dha) and dehydrobutyrine (Dhb) to their biologically active forms [12]. They have a preserved thioether ring containing the amino acids lanthionine and/or 3-methyl-lanthionine. This type of bacteriocin has binding specificity for bacterial cells and kills susceptible cells primarily through membrane pore formation with a strong and wide spectrum of antimicrobial activity against Gram-positive bacteria [13]. The best-studied lantibiotic, nisin produced by *Lactococcus lactis*, is employed as a food preservative in many countries. In addition, some oral bacteria have been shown to produce lantibiotics. *Streptococcus mutans*, which is the principal etiological agent of human dental caries [14], was shown to produce various kinds of lantibiotics such as Mutacin I, Mutacin II, Mutacin III (also known as Mutacin 1140), and Smb [15]. The biosynthetic loci of these lantibiotics are composed of multiple genes, including those involved in regulation, cleavage, transport, and immunity to the produced lantibiotics [16,17,18,19]. The bacteriocin- and immunity protein-encoding genes are generally co-transcribed to ensure that the producer strains are not killed by their own bacteriocin [20]. In addition, Smb is a two-component lantibiotic system and utilizes two peptides that are each posttranslationally modified to an active form and that act in synergy to produce antibacterial activity [19].

The lantibiotics show generally conserved characteristics such as resistance to acids, high temperature and digestive enzymes [21]. Most of the bacteriocins produced in the oral cavity are likely to be employed against neighboring oral bacteria. However, due to their resistance to the gastric environment, lantibiotics produced in the oral cavity potentially may be able to flow out into the intestinal tract maintaining a strong antimicrobial activity and as a result affect the intestinal microbiota. The main purpose of this study was to clarify the impact of lantibiotics produced by oral bacteria on the intestinal microbiota. We analyzed intestinal microbiota in children with few other causative factors for dysbiosis. We have found that the presence of lantibiotic bacteriocin-producing bacteria in the oral cavity tended to reduce the diversity of intestinal microbiota and significantly decreased the abundance of phylum Firmicutes in fecal microbiota.

## 2. Results

### 2.1. Overall In Vitro Susceptibility of Intestinal Microbiota Bacteria to Smb

Previous reports indicated that the lantibiotic bacteriocin Mutacin III, produced by *S. mutans*, has antimicrobial activity against bacteria in the intestinal microbiota, such as *Staphylococcus aureus*, *Clostridioides difficile*, *Enterococcus faecalis*, and *Enterococcus faecium* [22]. We were therefore interested in whether Smb and the other lantibiotics produced by *S. mutans* have antimicrobial activity against intestinal microbiota. To assess in vitro susceptibility of these Gram-positive bacteria to Smb, *S. mutans* Smb producing-reference strains GS5 and BM71 were used in a susceptibility assay against: *C. difficile*, *Clostridium perfringens*, *Finegoldia magna*, genus *Eubacterium* (*Eubacterium aerofaciens*, and *Eubacterium limosum*), genus *Enterococcus* (*E. faecalis* and *E. faecium*), and *S. aureus*. All of these indicators belong to the phylum Firmicutes. Although variations in the antimicrobial activities of both of these strains against these bacterial indicators were observed, all of these indicators except *E. faecium, E. faecalis*, and *S. aureus* were susceptible to Smb, while the Smb mutants of both strains were markedly attenuated in their ability to inhibit the growth of the indicator bacteria (Figure 1a). *E. faecium* exhibited weaker bacteriocin activity against both of wild type strains (Figure 1a). *E. faecalis* was not inhibited by either GS5 wild type or its mutant and BM71 wild type and mutants exhibited weaker bacteriocin activity. Neither wild type nor mutant strains exhibited inhibition of *S. aureus*. We also analyzed the in vitro susceptibility of genus *Bifidobacterium* which are Gram-positive bacteria and belong to phylum Actinobacteria to Smb. Strains GS5 and BM71 were assayed against *Bifidobacterium bifidum* and *Bifidobacterium breve*. Similar results to Firmicutes bacteria were observed (Figure 1b). Strong antibacterial activity was confirmed against *B. bifidum* and moderate activity was observed against *B. breve*. Conversely, there was no antibacterial activity to these bacteria in either of the mutant strains.

It has been reported that lantibiotics usually do not exert significant antimicrobial efficacy against intact Gram-negative bacteria [23]. Treatment with agents which can disrupt the outer lipopolysaccharide (LPS) rich membrane allows the bacteriocin to access the inner membrane and exert its antimicrobial effect [12]. However, some Gram-negative bacteria such as *Pectinatus frinsingensis*, an anaerobic microorganism responsible for spoilage of beer, do exhibit sensitivity to lantibiotics [24]. We assessed Smb sensitivity for *Klebsiella pneumoniae*, *Veillonela parvula*, and *Bacteroides fragilis*, which are major Gram-negative members of the intestinal microbiota. Smb producers were ineffective against all of these indicators (Figure 1c). Taken together, these results indicate that Smb, with some exceptions, has antibacterial activity against a large number of intestinal microbiome constituting Gram-positive bacteria.

### 2.2. Possible Impact of Lantibiotics Produced by Oral Bacteria on Intestinal Microbiota

The above results and previous paper [22] proved that Mutacin III and Smb have antibacterial activity against some intestinal Gram-positive bacteria. Mutacin I and Mutacin III have similar amino acid sequences and have been reported to have similar properties [15]. The lantibiotic Mutacin II has a different amino acid sequence and different properties to Mutacin I and Mutacin III [15,16]. Unfortunately, a Mutacin II producer was not available to us and the antimicrobial properties of Mutacin II against the bacteria constituting the intestinal microbiota are not clear. Mutacin II has been reported to exhibit very strong antimicrobial activity against some Streptococcus strains [25]. In order to investigate the influence of the lantibiotics Mutacin I, Mutacin II, Mutacin III, and Smb on intestinal microbiota, fecal samples were collected from 69 children aged 3 to 10 years attending a pediatric dental outpatient clinic (the details are shown in Materials and Methods). Firstly, we analyzed whether lantibiotic (Mutacin I, Mutacin II, Mutacin III, or Smb) -producing bacteria were present in saliva using nested PCR with each bacteriocin specific primer pair (Appendix A). Since Mutacin I and Mutacin III have a similar sequence, the primer pair used in this assay was the same (Mutacin I/III). Mutacin I/III was identified in 5 of 69 saliva specimens (a girl and four boys, an average age 8.4 ± 1.9 years, 5.0 or 9.25 ± 1.9 years for girl and boys, respectively) and Smb was identified in eight saliva specimens (two girls and six boys, an average age 5.6 ± 1.8 years, 6.5 ± 0.5 or 5.3 ± 0.8 years for girls and boys, respectively). We isolated two *S. mutans* strains harboring the Mutacin I gene cluster and performed antimicrobial testing using the bacteria described above. The results were similar to the Smb antimicrobial activity assay (Figure 1d–f). Therefore, Mutacin I was confirmed to have antimicrobial activity against intestinal microbiota constituting bacteria along with Smb and Mutacin III. Mutacin II-producing bacteria were not detected in any saliva samples. Based on these results, we divided the specimens into two groups, lantibiotic (Mutacin I/III and Smb) -producing bacteria positive group (group 1, 13 subjects; three girls and ten boys, an average age 6.7 ± 2.3 years, 6.0 ± 0.8 or 6.9 ± 2.5 years for girls and boys, respectively) and others (group 2, 56 subjects; sixteen girls and forty boys, an average age 6.4 ± 2.1 years, 6.2 ± 1.9 or 6.5 ± 2.2 years for girls and boys, respectively). Next, we examined the effect of acidity and temperature on the antimicrobial activity of Mutacin I and Smb. After cultivation of these producers stabbed on plates, diffused lantibiotics were treated with 0.01M hydrochloric acid (pH2) or heating at 60 °C. Inhibition zones against Streptococcus salivarius JCM5707 as an indicator were demonstrated, and antimicrobial activity of Smb and Mutacin I was not altered by these treatments (Figure 1g). 

### 2.3. Intestinal Microbial Composition Changes between Lantibiotic Positive and Negative Subjects

A total of 69 fecal samples from these subjects were analyzed using 16S ribosomal RNA gene sequencing. Species richness and evenness (alpha diversity) of the fecal microbiota were measured by Chao1 index based on Bray-Curtis and Shannon diversity index (Figure 2a,b). Interestingly, although we observed no significant differences between group 1 and group 2 in both alpha diversity indices (the statistical values of Chao1 and Shannon index by Mann-Whitney U-test were *p* = 0.08291 and *p* = 0.19740, respectively), the species richness in group 1 clearly showed a tendency to decrease (Chao1 result). Taking richness into account for evenness, there was a slightly decreasing trend (Shannon index result), suggesting that the lantibiotics produced by oral bacteria may influence intestinal microbiota. Unfortunately, we did not observe any significant difference by PERMANOVA comparing the distances between the groups in principal coordinate analysis (PCoA) based on UniFrac and Bray-Curtis dissimilarity (the weighted UniFrac: Appendix A, the unweighted UniFrac: Appendix A, and Bray-Curtis: Appendix A, PERMANOVA *p* = 0.98 for the weighted UniFrac, *p* = 0.511 for the unweighted UniFrac, or *p* = 0.826 for Bray-Curtis dissimilarity). We investigated taxonomic changes between lantibiotic positive and negative groups. The microbial community of both groups in the fecal microbiota were predominantly comprised of Firmicutes, Bacteroidetes, Proteobacteria, and Actinobacteria (Figure 2c and Appendix A), and the most dominant phylum in both groups was Firmicutes. However, the abundance of Firmicutes in the lantibiotic positive group (group 1) was significantly decreased compared to the lantibiotic negative group (group 2) (Figure 2d, Mann-Whitney U-test *p* = 0.04441). For Lentisphaerae, group 1 had a significantly increased abundance compared to group 2, but only one person in group 1 had a high proportion, with the rest having no value (Appendix A). For Proteobacteria, it seemed that the ratio in group 1 was slightly higher than that in group 2, but there was no significant difference (Figure 2c, Appendix A, and Appendix A), because only one person in group 1 had a relatively high proportion of Proteobacteria. No significant change was observed in either group for Actinobacteria or Bacteroidetes (Appendix A). To further confirm the decreased abundance of Firmicutes in group 1, we carried out quantitative real-time PCR with Firmicutes specific primer pairs (Table 1 and see Materials and Methods). The result was similar to the above metagenomic sequence data confirming that the ratio of Firmicutes in group 1 was significantly decreased compared to that in group 2 (Figure 2e). These results indicated that lantibiotics produced by oral bacteria may exhibit an antibacterial effect on Firmicutes. We investigated changes at genus level of group 1 showing that Genera Anaerostipes, ph2, Holdemania, and cc-115, were significantly depleted (Figure 2f–i, Mann-Whitney U-test *p* = 0.01508 for Anaerostipes, *p* = 0.02496 for ph2, *p* = 0.02446 for Holdemania, or *p* = 0.03814 for cc-115) compared to group 2.

### 2.4. Impact of Both Mutacin I/III and Smb on Intestinal Microbiota

To clarify which of Mutacin I/III or Smb most affect the intestinal microbiota, we divided the specimens into three groups (group 1a: n = 5, Mutacin I/III producing bacteria positive, group 1b: n = 8, Smb positive, and group 2: n = 56, lantibiotic negative) and analyzed the fecal microbiome composition. We compared the intestinal microbiota richness and evenness between the groups by investigating Chao1 and Shannon index of the alpha diversity. The tendency of decreased diversity in the lantibiotic positive groups (group 1a and group 1b) was again observed compared to the negative group (group 2), especially in species richness. However, it was not significant, and considering both richness and evenness, there was almost no difference between the groups (Appendix A, Kruskal-Wallis H test *p* = 0.228418 for Chao1, *p* = 0.22329 for Shannon index). In taxon-based analysis, no significant difference at phylum level was observed between the groups (Appendix A, and Appendix A). However, regarding Firmicutes, both lantibiotic positive groups 1a and 1b tended to show a decrease compared to the lantibiotic negative group 2 (Appendix A). At genus level, only the abundance of Anaerostipes in both lantibiotic positive groups showed a significant decrease (group 1a vs. group 2), or tendency to decrease (group 1b vs. group 2) compared to the negative group (Kruskal-Wallis H test *p* = 0.022874, Mann-Whitney U-test *p* = 0.0121 for group 1a vs. group 2) (Appendix A). In the family Ruminococcaceae (not decided at genus level), the proportion in the Mutacin I/III group (group 1a) was significantly decreased compared to the other groups (1b and 2) (Kruskal-Wallis H test *p* = 0.042866, Mann-Whitney U-test *p* = 0.0192 or 0.0167 for group 1a vs. group 1b or vs. group 2, respectively, Appendix A). On the other hand, in genus Bulleidia, the proportion in the Smb group (group 1b) was significantly decreased compared to group 1a and there was a tendency to decrease comparing group 1a to group 2 (Kruskal-Wallis H test *p* = 0.026845, Mann-Whitney U-test *p* = 0.0068 for group 1b vs. group 1a) (Appendix A). These results suggested that although there may be differences in the target bacteria, both Mutacin I/III and Smb can affect and reduce the relative abundance of Firmicutes. Genus Selenomonas was detected in more than half of patients in group 1a, but was hardly detected in other groups (Kruskal-Wallis H test *p* = 0.000032, Mann-Whitney U-test *p* = 0.0175, or *p* = 0.0001 for group1a vs. group 1b or group 1a vs. group 2, respectively) (Appendix A). The genus Selenomonas are Gram-negative bacteria belonging to the phylum Firmicutes, the class Clostoridia and the family Veillonellaceae. Thus, this result seems to be reasonable since lantibiotics usually do not exhibit antimicrobial activity against Gram-negative bacteria. The increase in the genus Selenomonas in group 1a may be due to a decrease in other antagonistic bacteria. It would therefore be interesting to investigate which kinds of bacteria antagonize the genus Selenomonas. Taken together, these results indicate that both Mutacin I/III and Smb may impact the intestinal microbiota.

### 2.5. Lantibiotics Produced by Oral Bacteria Do Not Affect Salivary Microbiota

As the antibacterial effects of lantibiotics produced by oral bacteria are likely to affect the oral microbiota, 16S rRNA gene profiling was carried out on saliva samples to examine whether the presence of Mutacin I/III-or Smb-producing bacteria alters the salivary microbial community. Salivary microbial alpha diversity of subjects analyzed with Chao1 and Shannon index demonstrate that there is no difference in species richness and evenness between groups (Appendix A, the statistical values of Chao1 and Shannon index by Mann-Whitney U-test were *p* = 0.97552 and *p* = 0.98776, respectively). In addition, Principal-coordinate analysis (PCoA) showed that there was no separation between these groups (the weighted UniFrac: Appendix A and the unweighted UniFrac: Appendix A and Bray-Curtis: Appendix A, the statistical values of the weighted UniFrac, unweighted UniFrac and Bray-Curtis by PERMANOVA were *p* = 0.141, *p* = 0.275 and *p* = 0.421, respectively). Based on the results of the taxon-based analysis for determination of the composition change between the groups, the abundance of the Firmicutes in group 1 seemed to be slightly decreased compared to group 2 (Appendix A); however, this change was not significant (Appendix A). Furthermore, a similar result was obtained with quantitative real-time PCR done with a Firmicutes specific primer pair (Appendix A). Similarly, the abundance of Proteobacteria in group 1 was only slightly increased compared to group 2 (Appendix A) and this change was also not significant. The abundances of Actinobacteria and Bacteroidetes were similar in both groups (Appendix A). Taken together, these results indicated that the lantibiotics produced in the oral cavity do not affect salivary microbiota at phylum level. At genus level, the lantibiotic positive group exhibited a significant decrease in the abundance of genus Porphyromonas (Appendix A) and [Prevotella] (Appendix A), which belong to phylum Bacteroidetes, and genus Dorea (Appendix A), belonging to phylum Firmicutes, family Lachnospiraceae (Mann-Whitney U-test *p* = 0.04126 for Porphyromonas, *p* = 0.01408 for [Prevotella], or *p* = 0.02497 for Dorea) compared to the lantibiotic negative group.

### 2.6. Administration of the Supernatant Containing Mutacin I Reduces Firmicute Bacteria in the Mouse Intestine

In order to determine whether lantibiotics can perturb mouse intestinal microbiota, the supernatant of a Mutacin I-producing *S. mutans* strain was orally administered to ICR mice twice a day for 4 days (group 3). As controls, the supernatants from either Mutacin I mutant *S. mutans* strain (group 4) or medium only (group 5) were also administered to mice (n = 10 per group). Species richness and evenness (alpha diversity) of the mice fecal microbiota were measured by Chao1 index based on Bray-Curtis and Shannon diversity index (Appendix A). No significant differences between the groups in both alpha diversity indices were observed (Kruskal-Wallis H test *p* = 0.09713 for Chao1 and *p* = 0.929091 for Shannon index). The species richness of mouse fecal microbiota in group 3 and group 4 exhibited a tendency to decrease compared to the control group (group 5) (Appendix A). The species evenness in addition to richness did not differ between the groups (Shannon index result, Appendix A). PCoA plots based on UniFrac and Bray-Curtis dissimilarity (the weighted UniFrac: Figure 3a, the unweighted UniFrac: Figure 3b, and Bray-Curtis: Figure 3c) appeared to be randomly distributed. However, PERMANOVA revealed significant composition differences between group 3 (lantibiotic containing culture supernatant) and group 4 (culture supernatant without lantibiotics) in all of the PCoA analysis results.

For taxon-based analysis, a total of nine phyla were detected (Figure 3d) and were listed in Appendix A. In the Mutacin I mouse group (group 3), Firmicutes tended to be decreased (Figure 3e) and Bacteroidetes also tended to be increased in the composition compared to group 4 and group 5 mice (Figure 3f). To confirm the abundance of Firmicutes, we carried out quantitative real-time PCR as previously detailed and the abundance of Firmicutes was found to be significantly decreased in group 3 compared to group 4 and group 5 (Kruskal-Wallis H test *p* = 0.017552, Mann-Whitney U-test *p* = 0.0413, or *p* = 0.0102 for group 3 vs. group 4 or group 3 vs. group 5, respectively) (Figure 3g). We investigated the changes at genus level between group 3 and other groups. Family Lachnospiraceae (not decided at genus level), which belongs to phylum Firmicutes, was significantly depleted in group 3 compared to other groups (Kruskal-Wallis H test *p* = 0.009908, Mann-Whitney U-test *p* = 0.0054, or *p* = 0.0154 for group 3 vs. group 4 or group 3 vs. group 5, respectively) (Appendix A). On the other hand, genus Prevotella, which belongs to Bacteroidetes, was significantly increased in group 3 compared to other groups (Kruskal-Wallis H test *p* = 0.000357, Mann-Whitney U-test *p* = 0.0032, or *p* = 0.0002 for group 3 vs. group 4 or group 3 vs. group 5, respectively) (Appendix A).

## 3. Discussion

Dysbiosis of intestinal microbiota can be caused by factors such as age, diet, infection, and drug consumption [11,26,27]. Amongst these, diet and drugs, especially antimicrobial agents, have the strongest effect on the intestinal microbial ecosystem [28]. Excess nutrients result in long-term changes to the intestinal microbiota, leading to decreasing microbial diversity, over-representation of Firmicutes, and a corresponding decrease in the proportion of Bacteroidetes [29,30]. Similarly, treatment of children with macrolides leads to long-term decreases in Firmicutes and Actinobacteria with concomitant increase in Bacteroidetes and Proteobacteria [31]. Intestinal microbiome dysbiosis subsequently leads to health disorders such as obesity, Type II diabetes, and inflammatory bowel disease [32,33]. Here, we show that colonization of the oral cavity by lantibiotic producing-bacteria results in decrease of bacterial richness and proportion of Firmicutes in the intestinal microbiota, suggesting that it may be a factor causing dysbiosis.

In this study, the lantibiotic positive subjects exhibited decreases in species richness of the intestinal microbiota compared to negative subjects, but the difference did not reach significance (Figure 2a,b). In addition, the abundance of Firmicutes was significantly decreased in the lantibiotic positive subjects compared to the negative subjects based on the results of taxon-based analysis and quantitative real-time PCR (Figure 2d,e). Slight decrease in the abundance of Actinobacteria was observed overall in the lantibiotic positive subjects (Appendix A). Since all the subjects that participated in this study were healthy and normally be considered a healthy control group, the differences in groupings between subjects are valuable findings, even though they were not significant. Firmicutes and Actinobacteria mainly consist of Gram-positive bacteria; therefore, it can be inferred that the reduction in abundance of these bacteria may be due to the influence of lantibiotics such as Mutacin I/III and Smb (Figure 1). This phenomenon was also confirmed in mice fed with the supernatant of Mutacin I-producing *S. mutans* (Figure 3d,f,g). Some members of the intestinal microbiome such as *Ruminocossus gnavis*, *Blautia obeum*, *E. faecalis*, *B. longum* and *L. lactis*, can produce lantibiotics [34] and these have been postulated to cause intestinal microbiota dysbiosis [35]. Although the lantibiotics produced by oral bacteria may impact on the intestinal microbiota, bacteriocins produced in the oral cavity had no significant impact on the salivary microbiota (Appendix A). We hypothesized that bacteria in the oral cavity may have resistance to endogenous antimicrobial agents produced in their environment and demonstrated that *E. faecium*, *E. faecalis*, and *S. aureus* which often colonize the oral cavity have lantibiotic resistance (Figure 1a,d). We can therefore speculate that lantibiotics produced by oral bacteria may have a strong influence on intestinal microbiota since they are produced distant to the intestinal tract. Further studies are required to address these issues.

Taxon-based analysis demonstrates the alteration in the intestinal microbiota of lantibiotic positive subjects compared to lantibiotic negative subjects (Figure 2c). Similar results were obtained by the analysis of fecal microbiota of mice fed with *S. mutans* culture supernatant containing lantibiotics (Figure 3d). The most typical change was a significantly decreased abundance of Firmicutes. In further extended analysis at genus level, there was a decrease in the proportion of Family Lachnospiraceae (decreases in *Anaerostipes* in human subjects and Family Lachnospiraceae in the mouse model) (Figure 2f and Appendix A). These bacteria produce butyrate [36] which is an important source of energy for colonic epithelial cells, enhances epithelial barrier integrity and modulates the gastrointestinal tract [37]. In addition, a decreased abundance of *Anaerostipes* was detected in patients with type 2 diabetes [38]. Further analysis to examine whether there is an association of the reduction of *Anaerostipes* by lantibiotics produced by oral bacteria in diabetic patients will be of interest to validate and expand the current findings.

*S. mutans* is the most typical oral lantibiotic producer. This bacterium causes human dental caries and a virulence property of this bacterium is its ability to form biofilm on tooth surfaces. In addition, *S. mutans* is known to be a possible pathogen for bacteremia and infection with *S. mutans* is a potential risk factor for cerebral haemorrhage [39]. Recently, this bacterium was noted to produce diverse families of molecules such as polyketide synthases, nonribosomal peptide synthetases, as well as the ribosomally synthesized and post-translationally modified peptides, such as lantibiotics [40]. The currently characterized lantibiotics produced by *S. mutans* include Mutacin K8 in addition to Mutacin I, Mutacin II, Mutacin III, and Smb [15] investigated in this study. The antibacterial actions and spectra of Mutacin I, Mutacin II, Mutacin III, and Smb have been well characterized [16,17,18,19]. On the other hand, there is only one report regarding Mutacin K8 [41] and its in-depth properties and antimicrobial activity target have not yet been clarified. We have isolated several *S. mutans* which harbor the Mutacin K8 gene cluster from salivary samples used in this study. However, these isolates did not demonstrate any antimicrobial activity against either intestinal microbiota or oral bacteria used in this study. We speculated that the gene producing the antimicrobial substance may not be expressed even though the Mutacin K8 gene cluster was present. As a result, we excluded Mutacin K8 from the lantibiotics group in this study. Furthermore, we previously demonstrated that differential antimicrobial activities of Smb in *S. mutans* is dependent on a point mutation in the flanking region of the promoter structure of *smbA* in vitro [42]. We assayed for the presence of the point mutation in all of the *smb* sequences from positive saliva samples used in this study, but did not find it. It is also unclear whether this is an exclusively in vitro effect or whether it also occurs in vivo in the human intestine. Furthermore, there are other bacteria in the oral cavity producing lantibiotics such as *S. salivarius* (Salivaricin) [43], and *E. faecalis* (Cytolysin) [44] which were not investigated in this study. Larger studies taking these factors into account are required to validate and expand the current findings. Nevertheless, this is the first demonstration for a relationship between oral bacteria and intestinal microbiota postulating a theory for induction of intestinal microbial dysbiosis.

## 4. Materials and Methods

### 4.1. Bacterial Strains and Agar Plate Bacteriocin Assay

*S. mutans* GS5 and BM71, which are Smb producer strains, were used in this study for the antimicrobial assay against intestinal microbiota. The derivative Δ*smbA* (GS5) and Δ*smbAB* (BM71) strains were constructed as previously described [19,42]. These strains were grown in Todd-Hewitt (TH) medium (Nippon Beckton Dickinson Co. Ltd., Tokyo, Japan) under anaerobic conditions. The mutant strains were grown in TH medium supplemented with 10 mg of erythromycin per mL *C. difficile* ATCC 9689, *C. perfringens* JCM3817, *F. magna* ATCC 29328, *E. aerofaciens* JCM10188, *E. limosum* JCM6421, *E. faecium* JCM5804, *E. faecalis* JCM5803, *S. aureus* ATCC 25923, *B. bifidum* JCM1255, *B. breve* JCM1192, *K. pneumoniae* ATCC 13883, *V. parvula* ATCC 10790, and *B. fragilis* ATCC 25285 were used as indicator strains for the Smb or Mutacin I activity assay. These bacterial strains were grown in Gifu anaerobic medium (GAM medium) (Nissui Pharmaceutical Co. Ltd. Tokyo, Japan) under anaerobic conditions. Agar plate bacteriocin assay was performed as previously described [19].

### 4.2. Fecal and Saliva Sample Collection

This study protocol was undertaken in accordance with the Declaration of Helsinki with approval by the Ethics Committee of Tokyo Medical and Dental University and Kyorin University (Tokyo Medical and Dental University IRB number: D2015-517, 13 May 2016 and Kyorin University IRB number: 813-01, 18 August 2016). After the aim and details of the experiments were explained, consent was obtained from all subjects prior to obtaining the samples. Fecal and saliva samples were collected from patients attending the Pediatric Dentistry Department, Tokyo Medical Dental University who were 19 girls and 50 boys, aged between 3 and 10 years with an average age of 6.4 years (S.D. = 2.1 years, 6.2 ± 1.9, or 6.5 ± 2.2 years for girls or boys, respectively). All volunteers had not received any medical treatment except dental care (e.g., caries treatment and periodic dental examination) and had not consumed medication including antibiotics within the last 2 weeks. Fecal samples were suspended in 1 mL of guanidine thiocyanate solution (100 mm Tris-HCl (pH 9.0), 40 mm EDTA and 4 m guanidine thiocyanate) [45]. Saliva samples were collected and immediately frozen at −80 °C until use.

### 4.3. Mouse Fecal Sample Collection

We purchased specific pathogen-free 5-week-old female ICR mice from CLEA Japan (CLEA Japan, Inc., Tokyo, Japan). Before the experiment, all mice were bred in different combinations three times for 2 weeks and when changing combinations, the bedding from all cages was mixed together and the mixed bedding was distributed to all cages. Each group of mice were maintained in separated plastic cages under standard laboratory conditions (room temperature 23 ± 2 °C, relative humidity 40–60%, 12h light-dark cycle) and fed with a standard diet (CE-2; Clea Japan) and sterilized tap water. The cell-free supernatant of a Mutacin I-producing *S. mutans* strain was orally administered by gavage to the mice twice a day for 4 days (group 3). As controls, the supernatants from either Mutacin I mutant *S. mutans* strain (group 4) or medium only (group 5) were also administered to mice (n = 10 per group). Fecal samples were obtained from mice on day 5 and immediately frozen at −80 °C until use. The experiments were approved by the Experimental Animal Ethics Committee of Kyorin University School of Medicine (approval No. 227).

### 4.4. Isolation of S. mutans and Construction of Mutants

*S. mutans* strains were isolated from the saliva of PCR positive samples by culturing on Mitis salivarius agar (Nippon Beckton Dickinson Co. Ltd., Tokyo, Japan) supplemented with 0.2 U/mL of bacitracin (MSB) [46]. After cultivation at 37 °C for 72 h under anaerobic conditions, isolates were identified by 16S rRNA sequences. Among the *S. mutans* isolates, two Mutacin I positive strains were used for construction of Mutacin I deficient mutants. The mutant with a defective *mutA* gene in the Mutacin I gene cluster was constructed by double-crossover homologous recombination via insertion of an erythromycin resistance determinant into the gene. The PCR fragments of the upstream and downstream regions of the gene were amplified with pairs of primers containing the BamHI site (MutI UF-MutI DRBam for upstream and MutI DFBam-MutI DR for downstream) (Table 1). After BamHI treatment, PCR products were ligated into pResEm10 plasmid [47] containing the Erm cassette. The ligation mixture served as a template for amplification of up-Em-down fragment with the upstream of Fw and downstream of Rev primers described above and the PCR product was used for transformation of *S. mutans* strains. Confirmation of gene disruption was determined by either PCR or DNA sequencing.

### 4.5. DNA Extraction from Fecal and Saliva Samples and 16S rRNA Sequences

Microbial DNA from fecal samples was extracted using QIAamp DNA Stool kit (Qiagen, Germantown, MD, USA) according to the manufacturer’s instructions with slight modification [48]. Briefly, lysis buffer containing fecal specimens was mixed with glass beads followed by bead beating three times for 30 s at a setting of 4200 rpm using a Multi-beads Shocker (MB755U, Yasui Kikaku, Tokyo, Japan). After 5 min incubation at 75 °C, the suspension was mixed again in the same manner. After centrifugation at 14,000 *g* for 5 min, subsequent steps were performed according to the manufacturer’s instructions. The DNA from saliva samples was extracted using a modified protocol with QIAamp DNA Mini Kit (Qiagen). Briefly, the harvested bacterial cells were suspended in 100 mL of Tris-EDTA buffer containing 3 mg/mL lysozyme and 40 U of mutanolysin (Sigma-Aldrich, St. Louis, MO, USA), and incubated at 37 °C for 1.5 h. The following steps were performed according to the manufacturer’s instructions. The DNA concentration was determined using a QuantiFluor dsDNA System and Quantus Fluorometer (Promega, Madison, WI, USA). In order to detect *S. mutans* in saliva subjects, nested PCR with *S. mutans* specific primers pairs was carried out as previously described [46].

### 4.6. Quantitative Real-Time PCR

Bacterial DNA from feces and saliva of the subjects or feces of male ICR mice was used for real-time-PCR using SYBR Premix Ex Taq (TAKARA Bio, Shiga, Japan). Bacterial identification was determined and Firmicutes-specific primer pairs (Firm934F and Firm1060R) were designed based upon the previous study [49] (Table 1). Quantitative data were calculated from a standard curve generated by amplifying serial dilutions of a known *S. mutans* DNA quantity of amplicon and the results were calculated as the abundance of the Firmicutes relative to that of all bacteria (Eub338F–Eub518R, Table 1). 

### 4.7. 16S rRNA Sequence Analysis

The V3-V4 region of the 16S rRNA gene was amplified from fecal, saliva, and mouse fecal samples using TAKARA Ex Taq Hot Start Version (TAKARA). The primers used for PCR amplification were 341F and 785R, which obtained Illumina index and sequencing adapter overhangs [50]. The amplicons generated from each sample were purified and selected by size using SPRIselect (Beckman Coulter, Brea, CA, USA). After determining the concentration of purified PCR products, equal amounts of the products were pooled. Sequencing was performed on an Illumina MiSeq sequencer with a MiSeq Reagent Kit V3 (Illumina, San Diego, CA, USA). Sequence processing and quality assessment were performed using open source software, the Quantitative Insights Into Microbial Ecology (QIIME) package (version 1.8.0) further) [51]. Row sequencing data were merged using the Biological Observation Matrix (BIOM) tables provided by QIIME into a unique *biom table* using a script included in the QIIME package. Pair-end reads were merged using the Fastq-join script in illumine-utils with the parameters *m* = 6 and *p* = 20, and then quality filtered using QIIME’s script *split_library_fastq.py* (*r* = 3, *p* = 0.75, *q* = 20, *n* = 0). De novo and reference-based chimerae detected by USERCH v6.1 with the Greengenes v13.8 database were removed. Sequences were clustered into operational taxonomic units (OTUs) based on 97% identity at the genus level using the UCLUST Algorithm [52] against the Greengenes database v13.8 database.

### 4.8. Statistical Analysis

Kruskal-Wallis and Mann-Whitney *U* tests were performed with Stat Flex software (ver. 6.0 Artech, Inc., Osaka, Japan). Permutational multivariate analysis of variance (PERMANOVA) was performed with R (“adonis” function in 3.6.3, vegan package [53]). Mann-Whitney U test was used to compare alpha diversity (Chao1 and Shannon diversity), and the change of the proportions of phyla and genera. The Kruskal-Wallis test was employed to examine QIIME generated bacterial abundance percentages to compare the abundance of each OUT when the number of groups was 3 or more. If the result of the Kruskal-Wallis test was significant, the difference between each group was determined by Mann-Whitney *U* test. PERMANOVA was used to assess the association with b-diversity measurement based upon distance matrices and permutation. Significance was assessed by 999 permutations and the covariate was adjusted. In all tests, *p* values less than 0.05 were considered statistically significant.

## 5. Conclusions

This study demonstrated that colonization of the oral cavity by lantibiotic producing-bacteria results in a decrease in bacterial species richness and proportion of Firmicutes in the intestinal microbiota. It can be inferred that the reduction in abundance of Firmicutes may be due to the influence of lantibiotics produced by oral bacteria. Since all the subjects that participated in this study were healthy and might normally be considered a control group, the differences in findings between subjects are useful data, even though they were not significant. The mechanisms for development of dysbiosis are still unclear, although some reports implicate exposure to various environmental factors. The results of this study suggest a role for lantibiotics as one of the factors contributing to dysbiosis. Further studies are in progress to examine the long-term effects of oral lantibiotics on the intestinal microbiota and larger scale studies are required to validate and expand the current findings.

## Figures and Tables

**Figure 1 ijms-22-03343-f001:**
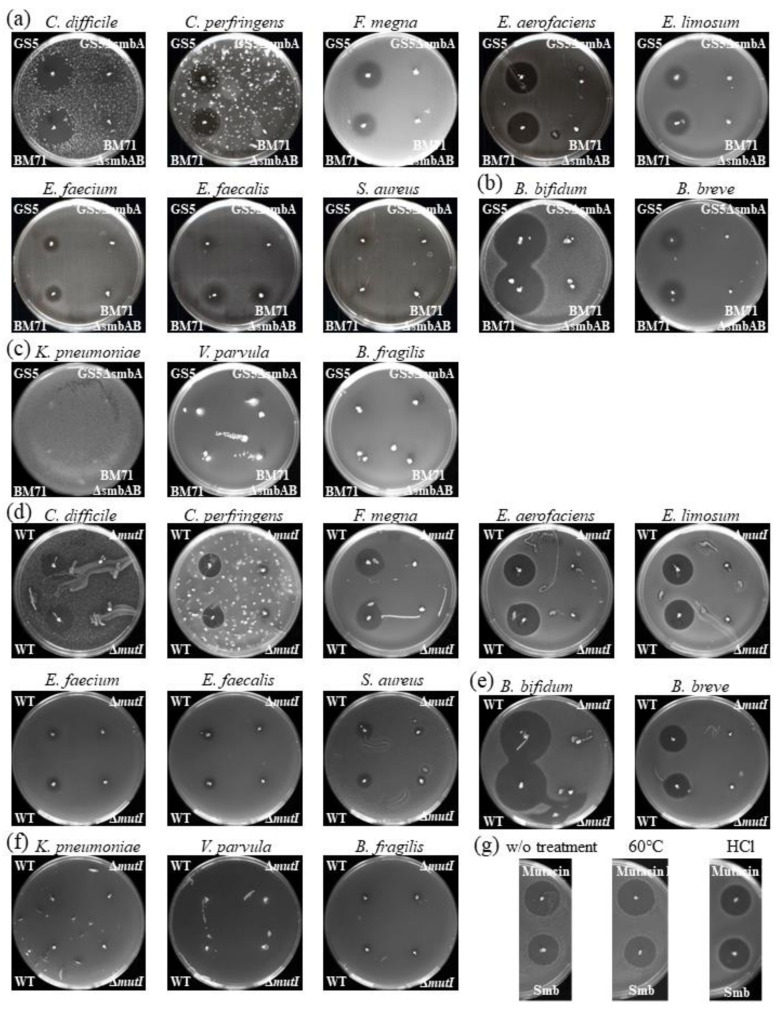
Results of agar plate bacteriocin assay. Smb activity of the producers (GS5, BM71 and its Smb mutant) against intestinal Gram-positive microbiota belonging to phylum Firmicutes (**a**), belonging to phylum Actinobacteria (**b**), or intestinal Gram-negative microbiota (**c**). Mutacin I activity of the clinical isolates against intestinal Gram-positive microbiota belonging to phylum Firmicutes (**d**), belonging to phylum Actinobacteria (**e**), or intestinal Gram-negative microbiota (**f**). Lantibiotic activity after treatment by heating (60 °C for 60 min), or 1N of HCl against *F. magna* (**g**).

**Figure 2 ijms-22-03343-f002:**
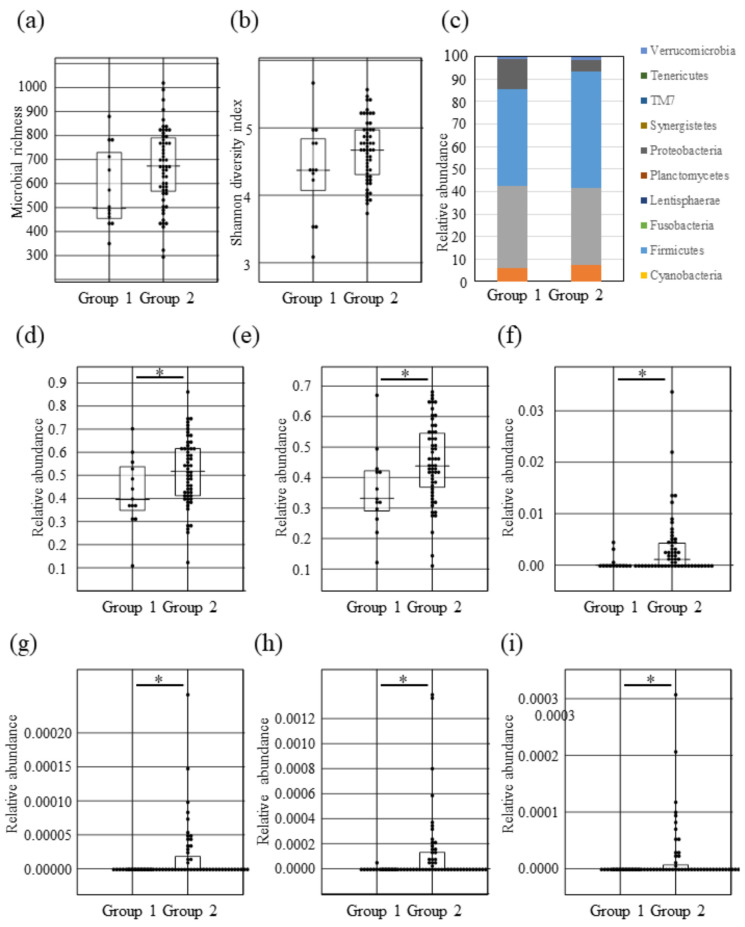
Comparison of the fecal microbial diversity and microbial taxonomic change of the fecal microbiota between the oral lantibiotic producer positive (group 1) and negative (group 2) subjects. Microbial richness (**a**) and Shannon diversity (**b**) based on operational taxonomic units (OTUs). The boxplots represent the diversity measures for 13 subjects (the lantibiotic positive group: Group 1) and 56 subjects (the lantibiotic negative group: Group 2). (**c**) Comparison of relative abundance of OTUs in bacterial composition of the fecal samples at phylum level between the lantibiotic positive group (Group 1; n = 13) and negative group (Group 2; n = 56). Relative abundance based on OTUs (**d**) or quantitative Real-time PCR (**e**) in Firmicutes of the fecal samples between groups. Relative abundance based on OTUs in genus Anaerostipes (**f**), ph2 (**g**), Holdemania (**h**), or cc-115 (**i**) in the fecal samples between groups. All of the boxplots for each group represent the interquartile range (25–75%) and the line within the box represents the median value. Mann-Whitney U-tests were used to test for significant differences between sample distances and asterisks show significant differences (*p* < 0.05).

**Figure 3 ijms-22-03343-f003:**
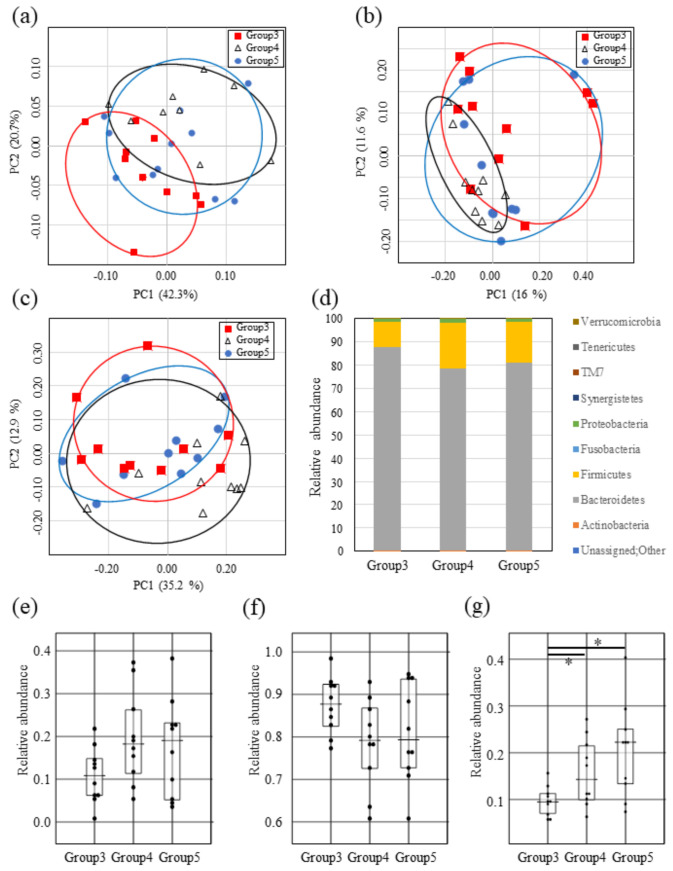
Comparison of the fecal microbial composition in mice fed with lantibiotics. Principal-coordinate analysis (PCoA) of mouse fecal microbiota fed with *S. mutans* supernatant containing Mutacin I (red circle), without Mutacin I (white triangle) or medium only (blue circle). PCoA plots were performed based on weighted UniFrac (**a**), unweighted UniFrac (**b**), or Bray-Curtis (**c**) distances of the mouse fecal bacterial communities. (**d**) Comparison of relative abundance of OTUs in the bacterial composition of the mouse fecal samples at phylum level between the *S. mutans* culture supernatant group (Group 3; n = 10), without lantibiotics group (Group 4; n = 10) and medium only group (Group 5; n = 10). (**e**) Relative abundance based on OTUs in Firmicutes of the mouse fecal samples between groups. Relative abundance based on OTUs (**f**) or quantitative Real-time PCR in Bacteroidetes of the mouse fecal samples between groups. (**g**) Relative abundance based on quantitative Real-time PCR in Firmicutes of the mouse fecal samples between groups. All of the boxplots for each group represent the interquartile range (25–75%) and the line within the box represents the median value. Kruskal-Wallis H-test was used to test for significant differences among sample distances and Mann-Whitney U-test was then used for significant differences between groups. Asterisks show significant differences (*p* < 0.05).

**Table 1 ijms-22-03343-t001:** Primers used in this study.

Primer.	Nucleotide Sequence (5′-3′)	Source of Reference
MutI/III F1st	GAGGCTAATGGTGGTATTAT	This study
MutI/III R1st	CCCACTTTACTATGAGTATC	This study
MutI/III F2nd	GTTTTCTGATATGCTTCTACTG	This study
MutI/III R2nd	CTAATATCAAAAGATTGTGCCG	This study
MutII F1st	GTGGTAAAAAAGATGGTAAACTG	This study
MutII R1st	TTAACAAGGTCCTGGTGGT	This study
MutII F2nd	ATGAACAAGTTAAACAGTAACGC	This study
MutII R2nd	CCGGTAAGTACATAGTGC	This study
Smb F1st	GCAATAACTTTTGGGTGGC	This study
Smb R1st	CCTTTATTTCCCAATACAATG	This study
Smb F2nd	GGAGCATTATGATGATAGGT	This study
Smb R2nd	TTCTTGCAAGCCTGCTTT	This study
Firm934F	GGAGYATGTGGTTTAATTCGAAGCA	49
Firm1060R	AGCTGACGACAACCATGCAC	49
Eub338F	AGCTGACGACAACCATGCAC	49
Eub518R	CGCTACTTGGCTGGTTCAG	49
MutI UF	GAAGAGTGGACTGAGTATG	This study
MutI URBam	CGGGATCCAGTATCTGTCCTCCTCGAA	This study
MutI DFBam	CGGGATCCCAAGGACTTCTAATAATTGTGTG	This study
MutI DR	GTTTAGAAACTTCTGTTTGACTATAC	This study

Restriction site sequences are underlined.

## Data Availability

The data that support the findings of this study are available from the corresponding author upon reasonable request.

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
