# Peer review of "Lantibiotics Produced by Oral Inhabitants as a Trigger for Dysbiosis of Human Intestinal Microbiota"

_ijms, 2021, doi:10.3390/ijms22073343_

Round 1

Reviewer 1 Report

Review of the manuscript ijms-1143379

  1. Original Submission

1.1. Recommendation

Minor revision.

  1. Comments to Author:

Overview and general recommendation:

In this present manuscript, the authors aim to demonstrate the connection between lantibiotic bacteriocins produced in the oral cavity and intestinal microbiota dysbiosis. The central theme of the article is exciting and has a high degree of novelty.

2.1. Major comments:

I suggest to the authors to revise the abstract to fit better to an experimental study. Two-thirds of the abstract provides theoretical information indicating a review manuscript, which can be confusing. Your research and/or methods are missing from the abstract. The abstract should be standing independently without a reader having to read the entire manuscript.

For less informed readers, a broader description of lantibiotics would be helpful.

In addition to the investigated effect in this study (intestinal microbiota dysbiosis), the lantibiotic compounds may also be beneficial and exploited in the future; see reference Kim, S.G., et al. Microbiota-derived lantibiotic restores resistance against vancomycin resistant Enterococcus. Nature 572, 665–669 (2019)”.

 Were all children's subjects treated for tooth decays? Wasn't a comparison between children treated for tooth decay and children without tooth decay?

On the other hand, extrapolating the results from experimental animals (mice) to humans is even more difficult.

What is the next step after demonstrating the appearance of intestinal microbiota dysbiosis related to oral lantibiotics?

2.2. Minor comments:

Please explain the Smb on first use in the abstract because it can be confused with an abbreviation for less knowledgeable readers; a parenthesis with (a dipeptide lantibiotic) may be useful.

The graphical representation of figures S3-1 (c) and S4-1(f) is not very intuitive. More explanations are needed in the manuscript or figures improvement.

Why were faecal samples obtained from mice on day 5?

Author Response

Dear Reviewer,

We would like to thank you and the reviewers for reading our manuscript carefully and for giving useful suggestions. We have revised the manuscript entitled “Lantibiotics produced by oral inhabitants as a trigger for dysbiosis of human intestinal microbiota” by Yonezawa et al. (ijms-1143379) on the basis of the reviewers’ comments. We have attempted to address the suggestions raised by the reviewers as follows:

Reviewer 1

In this present manuscript, the authors aim to demonstrate the connection between lantibiotic bacteriocins produced in the oral cavity and intestinal microbiota dysbiosis. The central theme of the article is exciting and has a high degree of novelty.

2.1. Major comments:

I suggest to the authors to revise the abstract to fit better to an experimental study. Two-thirds of the abstract provides theoretical information indicating a review manuscript, which can be confusing. Your research and/or methods are missing from the abstract. The abstract should be standing independently without a reader having to read the entire manuscript.

We agree with this comment. We modified more clearly and described our experimental results in “Abstract” section. (Lines 15 – 22, in Red)

For less informed readers, a broader description of lantibiotics would be helpful.

Thank you for the helpful comment. We modified some sentence (Lines 51 – 63, in Red) and added detailed explanation about lantibiotics (Lines 50 – 61, 66 – 68, in Red) in “Introduction” section so that readers can understand about lantibiotics.

In addition to the investigated effect in this study (intestinal microbiota dysbiosis), the lantibiotic compounds may also be beneficial and exploited in the future; see reference “Kim, S.G., et al. Microbiota-derived lantibiotic restores resistance against vancomycin resistant Enterococcus. Nature 572, 665–669 (2019)”.

Thank you for the useful information. We think that lantibiotics produced by intestinal bacteria are likely to affect the intestinal microbiome and so a study to examine the effect of intestinally produced lantibiotics on intestinal bacteria in addition to oral bacteria would be of great interest. We are keen to proceed with further research with reference to this paper.

 Were all children's subjects treated for tooth decays? Wasn't a comparison between children treated for tooth decay and children without tooth decay?

We apologize for the lack of explanation. The subjects included patients treated for caries and patients only receiving periodic dental examination. We added this explanation in the text. (Lines 496 – 499, in Red). In this study, we have not made a comparison between patients with or without tooth decay, but we think this idea is very interesting. The progress of tooth decay may correlate with the numbers of S. mutans in the oral cavity and therefore the concentration of lantibiotics and potentially the effect on the intestinal microbiota. Unfortunately, we did not acquire information about tooth decay for the subjects participating in this study but would consider investigating this in further studies.

On the other hand, extrapolating the results from experimental animals (mice) to humans is even more difficult.

We appreciate this comment. Indeed, there are no reports concerning whether and how feeding supernatant containing lantibiotics to animal models affects their intestinal microbiota. Improvements to the method are needed and are a subject of future study. On the other hand, the results obtained in this study were very similar to those in humans possessing lantibiotic producing bacteria in the oral cavity, especially with respect to the decrease of Firmicutes. We think these results are very significant and we have included them in the results of this manuscript.

What is the next step after demonstrating the appearance of intestinal microbiota dysbiosis related to oral lantibiotics?

We are very interested in researching the relationship between patients who are being treated for tooth caries and their corresponding intestinal microbiota as mentioned above. Moreover, we would like to investigate the impact of lantibiotics produced by intestinal bacteria in addition to oral bacteria on the intestinal microbiota. In addition, as stated in the text, research on the long-term effects of oral lantibiotics on the intestinal microbiota and larger scale studies are ongoing. (Lines 606 – 608, in Red).  

2.2. Minor comments:

Please explain the Smb on first use in the abstract because it can be confused with an abbreviation for less knowledgeable readers; a parenthesis with (a dipeptide lantibiotic) may be useful.

According to the comment, we added “a dipeptide lantibiotic” in parentheses (line 16, in Red).

The graphical representation of figures S3-1 (c) and S4-1(f) is not very intuitive. More explanations are needed in the manuscript or figures improvement.

According to the comment, we improved these figures to make it easier to understand to readers.

Why were faecal samples obtained from mice on day 5?

As mentioned above, no previous experiments of this nature have been recorded. With reference to the experiment on antibiotic administration, we decided to collect the fecal samples on day 5.

We read over the manuscript again and revised it so that it should be clearer to the reader. In addition, the manuscript has received additional review by a native English speaking microbiologist who specializes in editing of medical scientific papers.

We trust that the revised version of our paper will be suitable for publication in “International Journal of Molecular Sciences”.

Yours sincerely,

Hideo Yonezawa

Hideo Yonezawa, D.D.S., Ph.D.

Department of Infectious Diseases, Kyorin Univeristy School of Medicine

6-20-2 Shinkawa, Mitaka-shi, Tokyo, 181-8611

Japan

Tel: 81-422-47-5511

E-mail: yonezawa@ks.kyorin-u.ac.jp

Reviewer 2 Report

This paper investigates the impact of lantibiotic-producing bacteria on the intestinal microbiota in a satisfactory way. The work is interesting and well-written. However, I have some minor observations that are listed in the following lines.

  1. There are some typos and some bacterial names that should be italic. The work would benefit from close editing.
  2. Results: Avoid using references in the Results section. The results should describe the observations and findings of the current study only. Rewrite this section.
  3. Add a schematic diagram to summarize the experimental design.
  4. Since the study involves human volunteers, add the study’s IRB number and approvals to the materials and methods section.
  5. Line 454: Is there a reason for choosing this age group?
  6. Line 463: How long were the mice acclimated before treatment? Were the animals individually housed? What were the diet and light/dark cycle used?
  7. Line 465: does orally administered mean by oral gavage or in drinking water?
  8. Lines 583 and 584: add the approval dates.

Author Response

Dear Reviewer,

We would like to thank you and the reviewers for reading our manuscript carefully and for giving useful suggestions. We have revised the manuscript entitled “Lantibiotics produced by oral inhabitants as a trigger for dysbiosis of human intestinal microbiota” by Yonezawa et al. (ijms-1143379) on the basis of the reviewers’ comments. We have attempted to address the suggestions raised by the reviewers as follows:

Reviewer 2

This paper investigates the impact of lantibiotic-producing bacteria on the intestinal microbiota in a satisfactory way. The work is interesting and well-written. However, I have some minor observations that are listed in the following lines.

There are some typos and some bacterial names that should be italic. The work would benefit from close editing.

Thank you for the helpful comment. In addition, we found that Figure legend had a big mistake. Explanation of (g) was missing in Figure 3. We apologize for the many mistakes. We read over the manuscript and revised such mistakes.

Results: Avoid using references in the Results section. The results should describe the observations and findings of the current study only. Rewrite this section.

We agree with the reviewer’s comment. We also think that using references in the Result section should be avoided. On the other hand, we think that it is difficult for the reader to understand the overall of the manuscript if only the results are described in Result section, since this paper contains many results and figures and the story is complicated. We thought that the current form would lead to better reader understanding.

Add a schematic diagram to summarize the experimental design.

Thank you. In general, having a schematic diagram will make it easier for the reader to understand. However, none of the subjects who participated in this study were excluded, and the subjects were only divided into two groups. As the grouping was also very simple, we didn’t think that a further schematic would necessarily add to the comprehensibility of this manuscript.

Since the study involves human volunteers, add the study’s IRB number and approvals to the materials and methods section.

According to the comment, we added the IRB number and approval data in the Materials and Methods section. (Lines 490 – 491, in Red).

Line 454: Is there a reason for choosing this age group?

Some reports indicated that intestinal dysbiosis is related to age (Das et al, J Biosci 2019, Manges et al, The Journal of Infectious diseases 2010, Turnbaugh et al, Nature 2006, Forslund et al, Nature 2015, Qin et al, Nature 2012). We decided to investigate intestinal microbiota in children with no other causative factors of dysbiosis. This explanation is added in the introduction section. (Lines 76 – 77, in Red).

Line 463: How long were the mice acclimated before treatment? Were the animals individually housed? What were the diet and light/dark cycle used?

Before the experiment, all mice were bred in different combinations three times for 2 weeks. Each group of mice were maintained in separated plastic cages under standard laboratory conditions (room temperature 23 ± 2 ℃; relative humidity 40-60%, 12h light-dark cycle) and fed with a standard diet (CE-2; Clea Japan) and sterilized tap water. This explanation has been added in the Materials and Methods section. (Lines 505 – 510, in Red)

Line 465: does orally administered mean by oral gavage or in drinking water?

We administered the supernatant to mice by oral gavage. We added the explanation in the text. (Lines 510 – 512, in Red).

Lines 583 and 584: add the approval dates.

According to the comment, we added approval data in this section. (Lines 626 – 630, in Red).

We read over the manuscript again and revised it so that it should be clearer to the reader. In addition, the manuscript has received additional review by a native English speaking microbiologist who specializes in editing of medical scientific papers.

We trust that the revised version of our paper will be suitable for publication in “International Journal of Molecular Sciences”.

Yours sincerely,

Hideo Yonezawa

Hideo Yonezawa, D.D.S., Ph.D.

Department of Infectious Diseases, Kyorin Univeristy School of Medicine

6-20-2 Shinkawa, Mitaka-shi, Tokyo, 181-8611

Japan

Tel: 81-422-47-5511

E-mail: yonezawa@ks.kyorin-u.ac.jp

Reviewer 3 Report

This is very well-written manuscript, I have no suggestions.

Author Response

Dear Reviewer,

We would like to thank you and the reviewers for reading our manuscript carefully and for giving useful suggestions. We have revised the manuscript entitled “Lantibiotics produced by oral inhabitants as a trigger for dysbiosis of human intestinal microbiota” by Yonezawa et al. (ijms-1143379) on the basis of the reviewers’ comments. We have attempted to address the suggestions raised by the reviewers as follows:

Reviewer 3

Comments and Suggestions for Authors

This is very well-written manuscript, I have no suggestions.

Thank you for this kind comment. We would like to continue experiments to obtain more detailed data in this field.

We read over the manuscript again and revised it so that it should be clearer to the reader. In addition, the manuscript has received additional review by a native English speaking microbiologist who specializes in editing of medical scientific papers.

We trust that the revised version of our paper will be suitable for publication in “International Journal of Molecular Sciences”.

Yours sincerely,

Hideo Yonezawa

Hideo Yonezawa, D.D.S., Ph.D.

Department of Infectious Diseases, Kyorin Univeristy School of Medicine

6-20-2 Shinkawa, Mitaka-shi, Tokyo, 181-8611

Japan

Tel: 81-422-47-5511

E-mail: yonezawa@ks.kyorin-u.ac.jp
